# A High-Precision 3D Target Perception Algorithm Based on a Mobile RFID Reader and Double Tags

**Yaqin Xie** [1,*], **Tianyuan Gu** [1], **Di Zheng** [1], **Yu Zhang** [1] and **Hai Huan** [2]

1   School of Electronics and Information Engineering, Nanjing University of Information Science and Technology, Nanjing 210044, China; 20211249215@nuist.edu.cn (T.G.); 20211249307@nuist.edu.cn (D.Z.); zhangyu@nuist.edu.cn (Y.Z.)

2   School of Artificial Intelligence (School of Future Technology), Nanjing University of Information Science and Technology, Nanjing 210044, China; haihuan@nuist.edu.cn

*   Correspondence: xyq@nuist.edu.cn

**Abstract:** With the popularization of positioning technology, more and more industries have begun to pay attention to the application and demand of location information, and almost all industries can benefit from low-cost and high-precision location information. This paper introduces a novel three-dimensional (3D) low-cost, high-precision target perception algorithm that utilizes a Radio Frequency Identification (RFID) mobile reader and double tags. Initially, the Received Signal Strength (RSS) is employed to estimate the approximate position of the target along the length direction of the shelf. Additionally, double tags are affixed to the target, enabling the perception of its approximate height and depth through phase information measurements. Subsequently, the obtained rough position serves as an initial value for calibration using the proposed algorithm, allowing for the refinement of the target's length information relative to the shelf. Simulation results demonstrate the exceptional accuracy of the proposed method in perceiving the 3D position information of the target, achieving centimeter-level sensing accuracy.

**Keywords:** RFID; object perception; RSS; phase information

## 1. Introduction

Radio Frequency Identification (RFID) technology has found wide-ranging applications in intelligent identification, sports tracking, logistics, library management, and storage management [1–6]. However, current approaches often require multiple readers, resulting in high deployment costs [7,8], while the fixed position of the reader [9,10] limits its overall utility. Existing perception-based methods [11–13] mainly focus on two-dimensional (2D) location information, relying on human movement or environmental changes. Additionally, these methods exhibit limited accuracy in three-dimensional (3D) positioning. This paper addresses the critical challenge of achieving cost-effective and highly accurate 3D target positioning in RFID research.

Based on different RFID signal types, there are three distinct types: Received Signal Strength (RSS)-based method, a phase-based method, and a hybrid method that combines both RSS and phase information.

The commonly used location method, which is based on RSS measurement, utilizes the transmission loss model of electromagnetic waves to achieve positioning. In [14], NI et al. introduced the LANDMARC system as a pioneering approach for positioning. This system deploys a series of RFID tags in the target areas and relies on the reader's RSS measurements for localization. However, due to its susceptibility to environmental interference, RSS proves to be unstable, necessitating a higher density of reference tags for improved positioning accuracy. In [11], Wang et al. presented a method for localization that is based on changes in RSS time series as humans move. This approach requires the participation of pedestrians, while the position of the reader remains fixed. Similarly, ref. [12] proposes a

relative positioning method that utilizes RSS features. This method determines specific 2D positions by employing an RSS ranging model, eliminating the need for reference tags.

Converting the phase difference of RFID into a difference in distance and using it for localization is a popular research method [15–22]. By deploying two antennas and measuring the phase difference between the tag and the antennas, an average positioning accuracy of 12.8 cm was achieved by combining the distance difference [17]. Similarly, in reference [20], the authors proposed using the Kalman filter to process the image sequence captured by the camera to obtain the reader's trajectory. This trajectory is then combined with the phase difference measurement collected from the unknown location of the tag to estimate the 3D position of the tag. In the literature [21], the phase information is used to solve the nonlinear equation system by the least squares method, which requires a small amount of calculation and can meet the high real-time requirements of mobile positioning. The literature [22] considers the multipath problem in the actual scene, only selects the line-of-sight signal in the actual algorithm, and constructs a set of hyperbolas based on the virtual antenna to achieve positioning. In addition, the authors of [13] introduced the Mobile RF-robot Localization (MRL) system. In this method, a reader equipped with two vertically deployed antennas moves along a straight line in the warehouse channel to obtain the phase difference and time information of tags placed on the shelf. The geometric relationship between the antenna trajectory and the tags on the shelf is leveraged to achieve high 2D accuracy. However, when the 3D MRL system is used to determine the location of the target to be measured, the positioning accuracy in the y and z dimensions is low.

Combining phase and RSS [23–26] has emerged as a promising direction in RFID localization. In reference [23], the utilization of RSS information is employed to swiftly narrow down the potential area of the target tag, followed by the application of phase information to enhance the accuracy of its position estimation. Nevertheless, this approach demands a substantial deployment of readers and antenna arrays to achieve a finer grid, enabling centimeter-level positioning accuracy. In reference [25], RSS and PDOA (Phase Difference of Arrival) information are combined as joint fingerprint features, which are then utilized in a deep convolution neural network-based localization method. Although this technique outperforms traditional fingerprint recognition methods, it requires significant time and effort for offline data collection.

This paper proposes an RFID positioning algorithm that combines RSS and phase information measurements, taking advantage of the easy acquisition of RSS signals and the high accuracy of the RFID phase-based positioning method. The algorithm utilizes a mobile robot to carry the RFID reader, enabling positioning based on a single reader. To begin with, the algorithm captures the height and depth information of the target relative to the shelf by sensing the phase information of all the targets equipped with double tags on the shelf. Subsequently, leveraging the path loss model and measured RSS, the algorithm determines the position of the target relative to the shelf column. Furthermore, the estimated position in the aforementioned three dimensions is utilized as the initial value, followed by calibration using the proposed algorithm in this paper. This calibration process aims to obtain the 3D high-precision position information of the target on the shelf. The main contributions of this paper are summarized as follows:

- In the context of the shelf scenario, this paper introduces a double-tags phase model and presents a closed-form solution for accurately determining the height and depth of the target in relation to the shelf.
- This paper introduces a two-step positioning algorithm that combines RSS and signal phase for enhanced accuracy. In the first step, the algorithm utilizes the RSS peak information and the double-tags phase model to estimate the initial position. In the second step, the mobile RFID reader and double-tags (MRRDT) algorithm is employed for further calibration which achieves a final 3D position estimation error of approximately 4 cm.

- Compared to existing 3D positioning algorithms based on passive RFID tags, the proposed method offers higher positioning accuracy while maintaining lower computational complexity.

The rest of this paper is organized as follows. Section 2 describes the system design which includes an overview of the system architecture, the estimation of specific coarse positions, and the fine calibration of the x-dimension using the MRRDT algorithm. Simulation results and corresponding analysis are discussed in Section 3. Finally, Section 4 concludes the paper.

## 2. System Desgin

### 2.1. System Architecture

Aiming at the perception problem of the location information of items placed on a shelf, the system model built in this paper is shown in Figure 1. In this scenario, the communication between the antenna and the tag is more inclined to LOS (line-of-sight) transmission. Each target to be tested on the shelf is affixed with two passive RFID tags $(T_a, T_b)$, and the two tags are separated by a distance of $d_{a,b}$. According to the work of [22], to effectively eliminate phase ambiguity, the distance $d_{a,b}$ between two tags cannot exceed $\frac{1}{4}$ wavelength (that is to say, $d_{a,b} < \frac{1}{4}\lambda$).

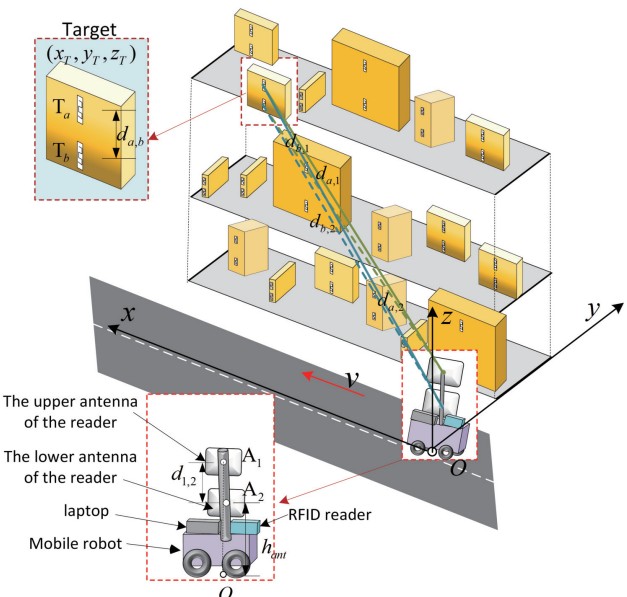

**Figure 1.** System architecture used by MRRDT.

Two tags appear in pairs for the same target to be tested, so they are named tag pairs. In addition, a mobile robot is placed on the horizontal ground, loaded with a portable computer and an RFID reader. The reader has two directional antennas, represented by $A_1$ and $A_2$. The two antennas are directly connected to the reader, and the two antennas face the shelf and are vertically placed on the robot platform. The distance between the center of the antenna is $d_{1,2}$, and the distance between the lower antenna and the ground is $h_{ant}$.

First, the mobile robot moves at a fixed speed $v$ from one end of the shelf to the other. Next, the reader carried by the mobile robot sends signals through the antenna to activate the passive tags pasted on the objects on the shelf and complete the backscattering. Then the dual antennas of the reader collect the backscattering signals of the tags. Finally, the portable computer stores and processes the collected signals and estimates the three-dimensional position information of the targets on the shelf according to the algorithm proposed in this paper.

At the beginning of robot motion, the projection of its center of gravity on the ground is the origin $O$ of coordinates. The motion direction of the robot is set as the *x*-axis, its

direction perpendicular to the ground is set as the $z$-axis, and the $y$-axis is perpendicular to the plane formed by $xoz$. According to the definition in Figure 1, the starting coordinates of the upper and lower antennas of the reader can be expressed as $(0, 0, h_{ant} + d_{1,2})$ and $(0, 0, h_{ant})$, respectively.

The three-dimensional high-precision target perception algorithm proposed in this paper is divided into two stages: rough estimation of three-dimensional target position and x-dimension position calibration. The specific flow of the algorithm is shown in Figure 2.

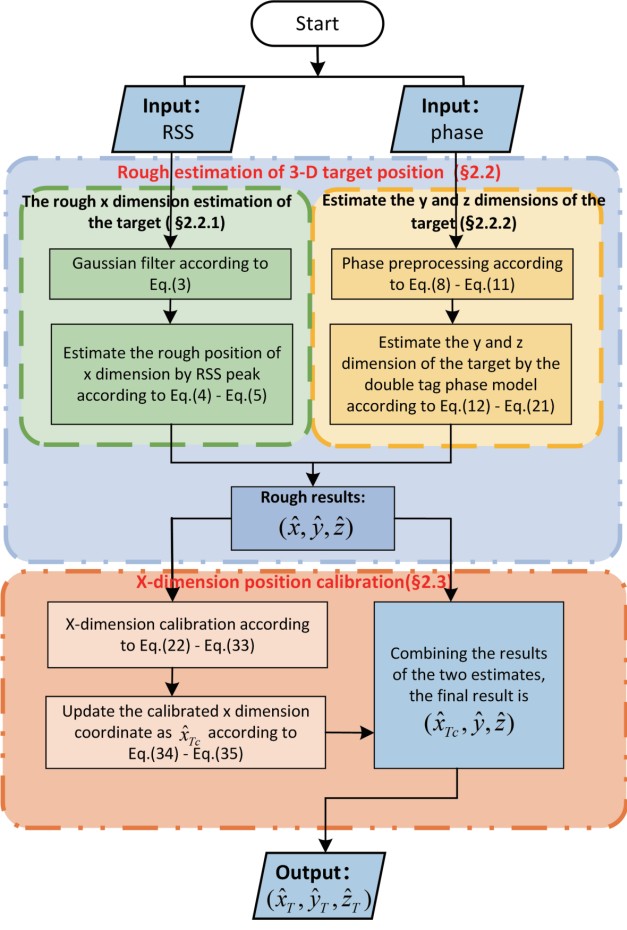

**Figure 2.** System architecture used by MRRDT.

### 2.2. Estimation of Rough Position

After the RFID reader collects a set of RSS and phase information, due to measurement error and noise, the RSS and phase information are filtered, respectively, at first. Then the coarse position of the target to be measured is estimated according to the filtered data.

#### 2.2.1. Perception of Rough Length

The total period of robot movement is sampled N times. Suppose the RSS value obtained by the tag $T_a$ and tag $T_b$ at time $t_n$ ($n \in [1, N]$) from the $i$th antenna ($i = 1,2$, respectively, correspond to the upper and lower antennas of the reader is $rss_{a,i}(t_n)$ and $rss_{b,i}(t_n)$. Then the total movement time of the robot can be obtained, and all the RSS values collected can be formed into the matrix as follows:

$$\mathbf{RSS} = \begin{bmatrix} \mathbf{rss}_{a,1} & \mathbf{rss}_{b,1} & \mathbf{rss}_{a,2} & \mathbf{rss}_{b,2} \end{bmatrix}^T \tag{1}$$

where

$$\mathbf{rss}_{g,i} = \begin{bmatrix} rss_{g,i}(t_1), \ldots, rss_{g,i}(t_n), \ldots, rss_{g,i}(t_N) \end{bmatrix}$$

where $g = a$ or $b$, correspond to the two tags pasted on the target, respectively.

The conclusion in [12] shows that each tag has a unique RSS profile, and we can use this uniqueness to distinguish different tags. Assume that the RSS received by the reader from tag $g$ at time $t_n$ is $rss_g(t_n)$, then $rss_{g,i}(t_n) = \frac{1}{S}\sum_{s=1}^{S} rss_{g,i}(t_{n,s})$, where $S$ represents the sampling times and $t_{n,s}$ represents the RSS obtained at the sampling point $s$ at time $t_n$. Define the RSS difference between tag a and tag b as $disRSS_{a,b}$:

$$disRSS_{a,b} = \sum_{n=1}^{N} |rss_a(t_n) - rss_b(t_n)| \tag{2}$$

In the actual environment, even if the RFID reader and the passive tag are both in a static state, the RSS values received by the reader at different times from the same tag are not constant. Noise or multipath effects in the communication environment may cause this situation. Take the RSS signal of the upper tag of the target to be tested received by the upper antenna of the reader at the fixed point as an example, as shown in the solid blue line in Figure 3 below. The initial signal fluctuated significantly, with the mean value of RSS being −41 dBm, the maximum value −37 dBm, and the minimum value −47 dBm, with a difference of 10 dBm between the maximum and minimum values.

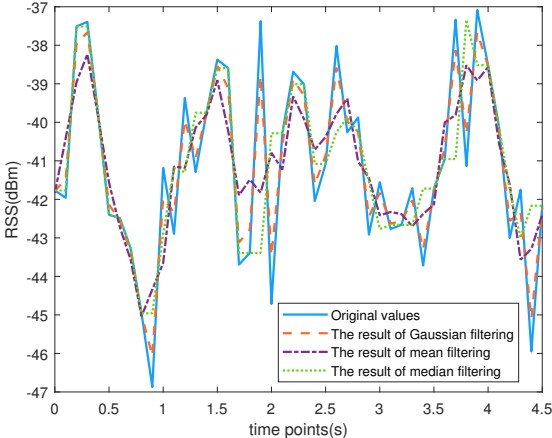

**Figure 3.** Changes in RSS measurements when the reader and tag position are fixed.

It can be seen that it is not necessarily accurate to directly use the mean value of these values in the case of a small amount of sampled data. Therefore, in this case, it is necessary to filter the RSS measurement signal, first remove the abnormal RSS measurement value, and then take the mean of other RSS values collected at the same time as the RSS value received by the antenna at that time from the tag to be tested.

Figure 3 also shows the effect of Gaussian filtering, mean filtering, and median filtering to remove outliers for the measured RSS values. It can be observed from the figure that the continuity of the signal obtained by Gaussian filtering is better and more details of the initial signal are preserved to a great extent. Therefore, Gaussian filtering is used to filter the original RSS measurements in the following sections.

First, Gaussian filtering is performed on the signal strength value $rss(t_n)$ collected at the time $t_n$ in the matrix shown in Equation (1). Then, the average value of multiple groups of values obtained simultaneously is taken to complete the smoothing processing. The processed **RSS** matrix can be expressed as:

$$\mathbf{RSS'} = \begin{bmatrix} \mathbf{rss'}_{a,1} & \mathbf{rss'}_{b,1} & \mathbf{rss'}_{a,2} & \mathbf{rss'}_{b,2} \end{bmatrix}^{\mathbf{T}} \tag{3}$$

where

$$\mathbf{rss'}_{g,i} = \begin{bmatrix} rss'_{g,i}(t_1), \dots, rss'_{g,i}(t_n), \dots, rss'_{g,i}(t_N) \end{bmatrix}$$

where $i = 1, 2$ , corresponds to the upper and lower antennas of the reader, and $g = a$ or $b$ correspond to the two tags pasted on the target, respectively. After observing the above pretreated RSS signal, it is found that the closer the antenna is to the tag, the stronger the RSS signal is. Therefore, the x-dimension coordinate of the tag under test can be estimated by finding the peak value of the processed RSS signal. For example, three tags are placed in positions $(0.45, 0.6, 0.7)$, $(1.56, 0.9, 1)$ and $(2.83, 0.8, 0.5)$, respectively, the unit is m, and the pre-processed RSS signal strength value is shown in Figure 4 below.

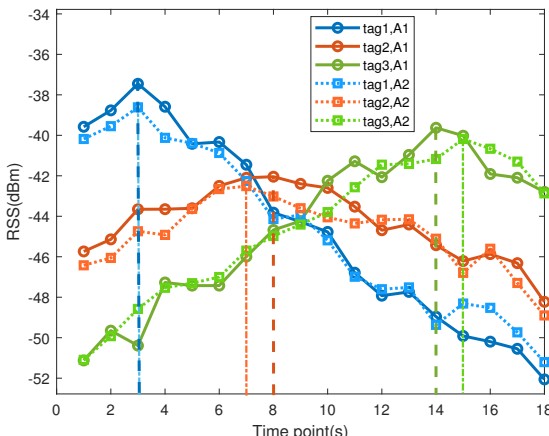

**Figure 4.** The RSS peak time corresponding to different tags.

It can be seen from this figure that for these three tags, the peak value of the RSS signal obtained by the upper antenna appears at 3 s, 8 s, and 14 s, respectively. The peak value of the RSS signal received by the lower antenna appeared at 3 s, 7 s, and 15 s, respectively. Therefore, the x-dimension coordinates corresponding to the three tags received by the antenna are, respectively, 0.6 m, 1.6 m, and 2.8 m. The x-dimension coordinates corresponding to the three tags obtained by the antenna are, respectively, 0.6 m, 1.4 m, and 3 m. To sum up, the x-dimension coordinates of the three tags corresponding to the RSS signal peak were 0.6 m, 1.5 m, and 2.9 m by averaging the above two groups of results.

Therefore, the peak value of **RSS**′ in Equation (3) is set as $t_{a,1}$, $t_{a,2}$, $t_{b,1}$ and $t_{b,2}$. Let $t_\perp$ represent the moment when the target is closest to the antenna, so

$$t_\perp = \frac{1}{4}(t_{a,1} + t_{b,1} + t_{a,2} + t_{b,2}) \tag{4}$$

Therefore, the estimated initial position value $\hat{x}$ of the target to be measured in the x-dimension can be expressed as

$$\hat{x} = v t_\perp \tag{5}$$

where $v$ represents the moving speed of the robot.

### 2.2.2. Perception of Rough Height and Depth

When the robot moves, the phase matrix $\tilde{\Phi}$ of the backscattered signal from the tag pair of the target to be sensed can be obtained

$$\tilde{\Phi} = \begin{bmatrix} \tilde{\phi}_{\mathbf{a,1}} & \tilde{\phi}_{\mathbf{b,1}} & \tilde{\phi}_{\mathbf{a,2}} & \tilde{\phi}_{\mathbf{b,2}} \end{bmatrix}^T \tag{6}$$

where

$$\tilde{\phi}_{g,i} = \begin{bmatrix} \tilde{\varphi}_{g,i}(t_1), \dots, \tilde{\varphi}_{g,i}(t_n), \dots, \tilde{\varphi}_{g,i}(t_N) \end{bmatrix}$$

where $i = 1, 2$ corresponds to the upper and lower antennas of the reader and $g = a$ or $b$ corresponds to the two tags pasted on the target, respectively. However, due to phase deflection caused by the multipath effect, environmental noise, and the difference between

equipment and tag, the phase of the tag's backscattered signal received by the antenna has interference, so that $\delta$ represents the phase deflection caused by environment or equipment. Thus, the phase acquired by the antenna $A_i$ of the reader at time $t_n$ can be expressed as

$$\tilde{\varphi}_{g,i}(t_n) = \text{mod}\left(\frac{4\pi d_{g,i}(t_n)}{\lambda} + \delta, 2\pi\right) \tag{7}$$

where $d_{g,i}(t_n)$ represents the distance between antenna $A_i$ and tag $T_g$ at time $t_n$. Since there is no jump between the continuous phase values, outliers can be found to compensate for whether the phase values jump in the adjacent moments so that the adjacent phase values are continuous. Similar to the unwrap command [27], the phase $\varphi_{g,i}^u(t_n)$ after compensation can be expressed as

$$\varphi_{g,i}^u(t_n) = \tilde{\varphi}_{g,i}(t_n) + 2k\pi \tag{8}$$

Due to the ambiguity of phase $\tilde{\varphi}_{g,i}(t_n)$ in Equation (8), and the $k$ value periodic element is used to restore the true phase. The phase compensation is carried out for $N$ groups of phases collected during the total movement time of the robot, and the phase matrix obtained can be expressed as

$$\Phi^u = \begin{bmatrix} \phi_{a,1}^u & \phi_{b,1}^u & \phi_{a,2}^u & \phi_{b,2}^u \end{bmatrix}^T \tag{9}$$

where

$$\phi_{g,i}^u = \begin{bmatrix} \varphi_{g,i}^u(t_1), \ldots, \varphi_{g,i}^u(t_n), \ldots, \varphi_{g,i}^u(t_N) \end{bmatrix}$$

where $i = 1, 2$ corresponds to the upper and lower antennas of the reader and $g = a$ or $b$ corresponds to the two tags pasted on the target, respectively.

In addition to the change in the distance leading to the change in phase, the phase deflection $\delta$ caused by other factors has not been eliminated. Therefore, after the phase unwrapping, this paper carries out phase value calibration similar to that in [28]. A tag with a known position was selected as a reference tag. A mobile robot with a single step length of 5 cm was used to measure the phase of the backscattered signal of the tag. Each measurement lasted for 30 s. At time $t_n$, the phase measurement value $\varphi_{g,i}^{test}(t_n)$ was unwrapped to obtain $\varphi_{g,i}^{u,test}(t_n)$, and the difference value $C_{g,i}(t_n)$ between the theoretical value of phase $\varphi_{g,i}^{u,theroy}(t_n)$ and the test value $\varphi_{g,i}^{u,test}(t_n)$ after phase unwrapping is taken as the phase correction value,

$$C_{g,i}(t_n) = \varphi_{g,i}^{u,theroy}(t_n) - \varphi_{g,i}^{u,test}(t_n) \tag{10}$$

Based on the correction value $C_{g,i}(t_n)$ shown in the above equation, the corrected phase $\phi_{g,i}^{u'}$ can be obtained by further modifying Equation (9).

$$\begin{aligned} \phi_{g,i}^{u'} &= \phi_{g,i}^u - [C_{g,i}(t_1), \ldots, C_{g,i}(t_n), \ldots, C_{g,i}(t_N)] \\ &= [\varphi_{g,i}^{u'}(t_1), \ldots, \varphi_{g,i}^{u'}(t_n), \ldots, \varphi_{g,i}^{u'}(t_N)] \end{aligned} \tag{11}$$

where $i = 1, 2$, corresponds to the upper and lower antennas of the reader and $g = a$ or $b$ correspond to the two tags pasted on the target, respectively.

After the phase matrix preprocessing is completed, because it is difficult to directly estimate the position of the target to be measured in any dimension using a single tag [11,12], this paper introduced the double-tags phase model algorithm to complete the position estimation of the two dimensions. The double-tags phase model is shown in Figure 5. As can be seen from this figure, the target to be tested is, respectively, pasted with the upper and lower tags $T_a$ and $T_b$, and the distance between the two tags is $d_{a,b}$. $d_{a,i}(t_n)$ and $d_{b,i}(t_n)$,

respectively, indicate the distance between the *i*th antenna of the reader and tags $T_a$ and $T_b$ at time $t_n$, then

$$d_{g,i}(t_n) = \frac{\lambda \varphi_{g,i}^{u'}(t_n)}{4\pi} \tag{12}$$

where $g = a$ or $b$, $\varphi_{a,i}^{u'}(t_n)$ and $\varphi_{b,i}^{u'}(t_n)$, respectively, represent the phase calibration values between the tag $T_a$, $T_b$ and the *i*th antenna of the reader at time $t_n$. $\lambda$ represents the wavelength corresponding to the electromagnetic wave emitted by the reader. Therefore, the distance vector $d_{a,i}$ and $d_{b,i}$ corresponding to $t_n$ of antenna $A_i$ and tags $T_a$ and $T_b$ at each time in the whole process can be expressed as

$$\mathbf{d}_{g,i} = [d_{g,i}(t_1), \dots, d_{g,i}(t_n), \dots, d_{g,i}(t_N)] \tag{13}$$

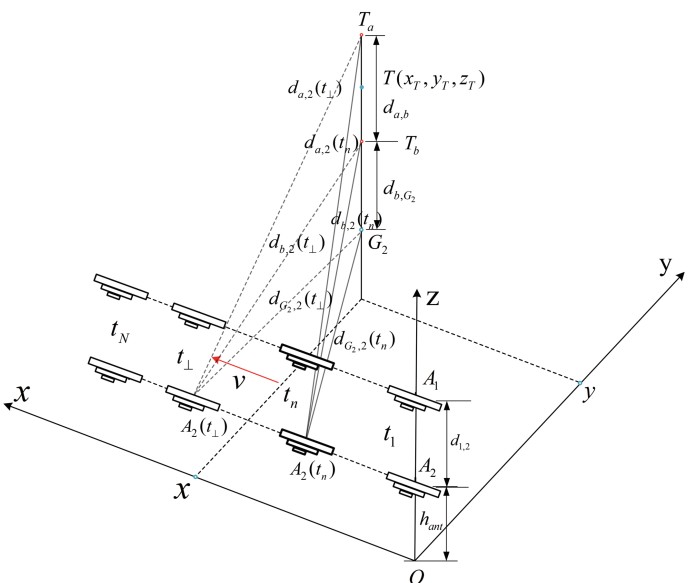

**Figure 5.** Double-tags phase model.

In Figure 5, at time $t_n$, the position of $A_i$ is $A_i(t_n)$, and the crossing point at is the vertical line of the tag $T_a$ and $T_b$, which intersects with the point $G_i$. Then tag $T_a$, $A_i(t_n)$ and $G_i$ can form a right triangle, and tag $T_b$, $A_i(t_n)$ and $G_i$ can also form a right triangle, so the time $t_n$ satisfies

$$d_{a,i}^2(t_n) - (d_{a,b} + d_{b,G_i}(t_n))^2 = d_{b,i}^2(t_n) - d_{b,G_i}^2(t_n) \tag{14}$$

where $d_{b,G_i}(t_n)$ represents the distance between tag $T_b$ and projection point $G_i$ at time $t_n$. According to Equation (14), we can obtain

$$d_{b,G_i}(t_n) = \frac{d^2{}_{a,i}(t_n) - d^2{}_{b,i}(t_n) - d^2{}_{a,b}}{2d_{a,b}} \tag{15}$$

According to the position relation of the target to be tested in Figure 5, it can be concluded that the estimated $z_i(t_n)$ of the target's z-dimension position at this time satisfies

$$z_i(t_n) = d_{b,G_i}(t_n) + \frac{1}{2}d_{a,b} + h_{ant} \tag{16}$$

By substituting $d_{b,G_i}(t_n)$ obtained from Equation (15) into Equation (16), $z_i(t_n)$ can be obtained. Then, average the estimated value of z-dimension $z_i$ of the target to be measured at all times, to satisfy

$$\hat{z}_i = \frac{1}{N}\sum_{n=1}^{N} z_i(t_n) \tag{17}$$

Meanwhile, in the right triangle formed by the tag $T_b$ and $A_i(t_n)$ and $G_i$, to satisfy

$$d_{G_i,i}(t_n) = \sqrt{d_{b,i}^2(t_n) - d_{b,G_i}^2(t_n)} \qquad (18)$$

where $d_{G_i,i}(t_n)$ is the distance between point $G_i$ at time $t_n$ and antenna $A_i(t_n)$. The $d_{G_i,i}(t_n)$ can be solved by substituting Equation (15). However, the true position of the y-dimension can only be obtained when the antenna is perpendicular to the tag time $t_\perp$, and the actual y value is $d_{G_i,i}(t_\perp)$. Now combing with the whole process to solve the corresponding y value at time $t_n$, to satisfy

$$y_i(t_n) = \sqrt{d_{G_i,i}^2(t_n) - [v(t_n - t_\perp)]^2} \qquad (19)$$

where $t_\perp$ can be obtained by Equation (4), and $v$ represents the moving speed of the robot. Then calculate the average $\hat{y}_i$ of the whole process by $y(t_n)$ in Equation (19), and $t_n$ belongs to $t_1$ through $t_N$, to satisfy:

$$\hat{y}_i = \frac{1}{N} \sum_{n=1}^{N} y_i(t_n) \qquad (20)$$

The double-tags phase model is described by a single antenna, but to obtain more accurate results, the average value of the two groups of $\hat{y}_i$ and $\hat{z}_i$ ( $i$ = 1, 2, respectively, correspond to the upper and lower antennas) obtained by the antenna $A_1$ and $A_2$ is obtained

$$\begin{cases} \hat{y} = \frac{1}{2}(\hat{y}_1 + \hat{y}_2) \\ \hat{z} = \frac{1}{2}(\hat{z}_1 + \hat{z}_2) \end{cases} \qquad (21)$$

By combining Equation (5) and Equation (21), the rough estimate of the initial position of the target to be measured can be updated as $\mathrm{MRRDT}_{first}$ and its result is $(\hat{x}, \hat{y}, \hat{z})$.

### 2.3. Calibration for Refinement

Due to the interference and noise in the signal measurement process, the above initial rough position estimation based on the measured value of the RSS signal has a large error. Thus, an x-dimension calibration algorithm based on the Taylor series expansion is proposed in this study to correct the target's length information relative to the shelf as shown in Equation (5), therefore enhancing the accuracy of the 3D position estimation. The detailed calibration algorithm is shown in Figure 6.

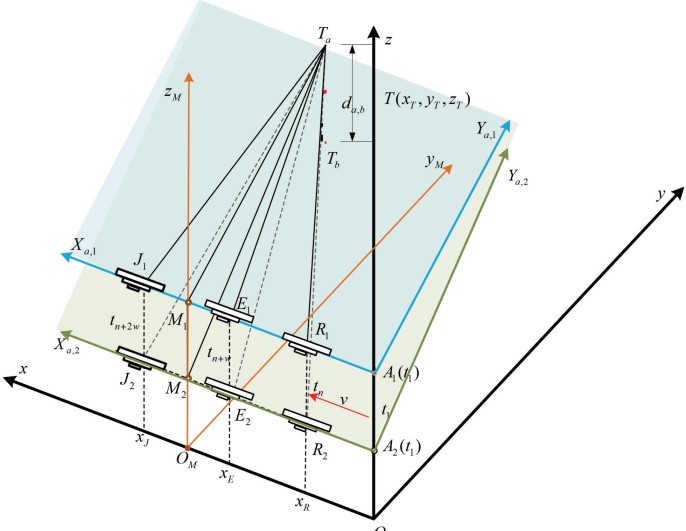

**Figure 6.** Diagram of 3D calibration algorithm.

As can be seen from Figure 6, four different color lines are used to form four different coordinate systems, respectively. The black bold coordinate systems have the same layout

as the original 3-D coordinate system in Figure 1. The orange one represents the 2-D coordinate system at the shortest distance between the antenna and the target. And the blue and green coordinate systems are the 2-D coordinate systems based on the plane formed by the origin and tag $T_a$ at the initial time of the two antennas respectively. The whole time process $t_N$ is evenly divided into three equally parts, namely $[t_1, \ldots, t_w]$, $[t_{w+1}, \ldots, t_{2w}]$ and $[t_{2w+1}, \ldots, t_N]$, where $N = 3w$. Let the position of antenna $A_1$ corresponding to $t_n$, $t_{n+w}$, and $t_{n+2w}$ be $R_1$, $E_1$ and $J_1$, respectively, and the position of antenna $A_2$ corresponding to the same time be $R_2$, $E_2$ and $J_2$, respectively. The calibration algorithm is used to calibrate the 3D rough estimation of the measured target. The algorithm requires at least one tag and two antennas of the reader. Taking tag $T_a$ as an example, tag $T_a$, $R_1$, $E_1$ and $J_1$ forms a plane, as shown in Figure 7.

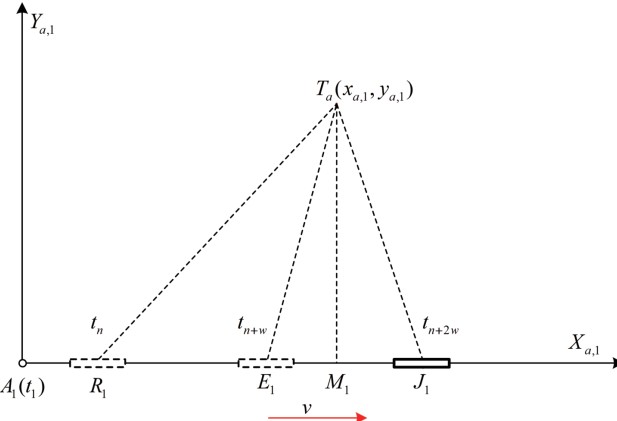

**Figure 7.** A two-dimensional plan of the calibration algorithm.

As shown in Figure 7, let the projection point of tag $T_a$ on the line where the three points of $R_1$, $E_1$ and $J_1$ are located be $M_1$. From the corresponding relation between phase and distance of backscattering in tag $T_a$ at different times, we can obtain

$$
\begin{cases}
\Delta\varphi_1 = \varphi_{a,1}^{u\prime}(t_n) - \varphi_{a,1}^{u\prime}(t_{n+w}) \\
\quad = \dfrac{4\pi}{\lambda}(\left|\overrightarrow{T_a R_1}\right| - \left|\overrightarrow{T_a E_1}\right|) \\
\Delta\varphi_2 = \varphi_{a,1}^{u\prime}(t_{n+w}) - \varphi_{a,1}^{u\prime}(t_{n+2w}) \\
\quad = \dfrac{4\pi}{\lambda}(\left|\overrightarrow{T_a E_1}\right| - \left|\overrightarrow{T_a J_1}\right|)
\end{cases}
\tag{22}
$$

In addition, the three points of $R_1$, $E_1$ and $J_1$ and the two points of tag $T_a$ and $M_1$, respectively, form three right triangles $Rt_{R_1 T_a M_1}$, $Rt_{E_1 T_a M_1}$ and $Rt_{J_1 T_a M_1}$, so as to satisfy

$$
\begin{cases}
\overrightarrow{T_a R_1} = \overrightarrow{T_a M_1} + \overrightarrow{M_1 R_1} \\
\overrightarrow{T_a E_1} = \overrightarrow{T_a M_1} + \overrightarrow{M_1 E_1} \\
\overrightarrow{T_a J_1} = \overrightarrow{T_a M_1} + \overrightarrow{M_1 J_1}
\end{cases}
\tag{23}
$$

In the $X_{a,1} A_1(t_1) Y_{a,1}$ plane, the position of tag $T_a$ can be expressed as $(x_{a,1}, y_{a,1})$. Combined with the position relation of the tag in the coordinate system of Figures 6 and 7, we can obtain

$$
\begin{cases}
\left|\overrightarrow{A_1(t_1) M_1}\right| = x_{a,1} \\
\left|\overrightarrow{T_a M_1}\right| = y_{a,1}
\end{cases}
\tag{24}
$$

In addition, since the velocity $v$ and time points $t_n$, $t_{n+w}$ and $t_{n+2w}$ are known, we can obtain

$$\begin{cases} \left|\overrightarrow{A_1(t_1)R_1}\right| = vt_n \\ \left|\overrightarrow{A_1(t_1)E_1}\right| = vt_{n+w} \\ \left|\overrightarrow{A_1(t_1)J_1}\right| = vt_{n+2w} \end{cases} \tag{25}$$

According to the geometric relationship shown in Figure 7 and combining Equations (24) and (25), we can obtain

$$\begin{cases} \left|\overrightarrow{M_1R_1}\right| = \left|\overrightarrow{A_1(t_1)M_1}\right| - \left|\overrightarrow{A_1(t_1)R_1}\right| \\ \left|\overrightarrow{M_1E_1}\right| = \left|\overrightarrow{A_1(t_1)M_1}\right| - \left|\overrightarrow{A_1(t_1)E_1}\right| \\ \left|\overrightarrow{M_1J_1}\right| = \left|\overrightarrow{A_1(t_1)M_1}\right| - \left|\overrightarrow{A_1(t_1)J_1}\right| \end{cases} \tag{26}$$

According to Equations (23) and (26), Equation (22) can be rewritten as

$$\begin{cases} \dfrac{\lambda}{4\pi}\Delta\varphi_1 = \sqrt{(x_{a,1} - v \cdot t_n)^2 + y_{a,1}^2} \\ \qquad\qquad - \sqrt{\left(x_{a,1} - v \cdot t_{n+w}\right)^2 + y_{a,1}^2} \\ \dfrac{\lambda}{4\pi}\Delta\varphi_2 = \sqrt{\left(x_{a,1} - v \cdot t_{n+w}\right)^2 + y_{a,1}^2} \\ \qquad\qquad - \sqrt{\left(x_{a,1} - v \cdot t_{n+2w}\right)^2 + y_{a,1}^2} \end{cases} \tag{27}$$

Since it is difficult to directly solve the nonlinear equations shown in the above equation, Taylor series expansion is carried out on the equation, and the rough estimate $(\hat{x}, \hat{y})$ of the x-dimension and y-dimension of the target to be measured obtained by Equation (5) and Equation (21) is taken as the initial value, which can be obtained:

$$AX = B \tag{28}$$

where

$$A = \begin{bmatrix} \dfrac{\hat{x}-vt_n}{\left(\left|\overrightarrow{T_aR_1}\right|\right)_0} - \dfrac{\hat{x}-vt_{n+w}}{\left(\left|\overrightarrow{T_aE_1}\right|\right)_0} & \dfrac{\hat{y}}{\left(\left|\overrightarrow{T_aR_1}\right|\right)_0} - \dfrac{\hat{y}}{\left(\left|\overrightarrow{T_aE_1}\right|\right)_0} \\ \dfrac{\hat{x}-vt_{n+w}}{\left(\left|\overrightarrow{T_aE_1}\right|\right)_0} - \dfrac{\hat{x}-vt_{n+2w}}{\left(\left|\overrightarrow{T_aJ_1}\right|\right)_0} & \dfrac{\hat{x}}{\left(\left|\overrightarrow{T_aE_1}\right|\right)_0} - \dfrac{\hat{x}}{\left(\left|\overrightarrow{T_aJ_1}\right|\right)_0} \end{bmatrix}$$

$$X = \begin{bmatrix} x_{a,1} & y_{a,1} \end{bmatrix}^T$$

$$B = \begin{bmatrix} \left(\sqrt{\left(\left|\overrightarrow{T_aE_1}\right|\right)_0} - \sqrt{\left(\left|\overrightarrow{T_aR_1}\right|\right)_0}\right) + \frac{\lambda}{4\pi}\Delta\varphi_1 + A_{11}\hat{x} + A_{12}\hat{y} \\ \left(\sqrt{\left(\left|\overrightarrow{T_aJ_1}\right|\right)_0} - \sqrt{\left(\left|\overrightarrow{T_aE_1}\right|\right)_0}\right) + \frac{\lambda}{4\pi}\Delta\varphi_2 + A_{21}\hat{x} + A_{22}\hat{y} \end{bmatrix}$$

and $\left(\left|\overrightarrow{T_aR_1}\right|\right)_0$, $\left(\left|\overrightarrow{T_aE_1}\right|\right)_0$ and $\left(\left|\overrightarrow{T_aJ_1}\right|\right)_0$ represent the value obtained by substituting the initial values of the estimated tags in Equations (5) and (21) into Equation (23), respectively. That is to say, $\left(\left|\overrightarrow{T_aR_1}\right|\right)_0 = \sqrt{(\hat{x} - vt_n)^2 + \hat{y}^2}$, $\left(\left|\overrightarrow{T_aE_1}\right|\right)_0 = \sqrt{(\hat{x} - vt_{n+w})^2 + \hat{y}^2}$ and $\left(\left|\overrightarrow{T_aJ_1}\right|\right)_0 = \sqrt{(\hat{x} - vt_{n+2w})^2 + \hat{y}^2}$.

So far, according to Equation (28), the $X_{a,1}$ axis and $Y_{a,1}$ axis projections of tag $T_a$ on this plane are $x_{a,1}$ and $y_{a,1}$, respectively, corresponding to $\left|\overrightarrow{A_1(t_1)M_1}\right|$ and $\left|\overrightarrow{T_aM_1}\right|$ respectively with

$$X = (A^T A)^{-1} A^T B \tag{29}$$

For Figure 6, $\left|\overrightarrow{A_1(t_1)M_1}\right|$ is the real x-axis position of tag $T_a$ of the target to be tested, while $\left|\overrightarrow{T_aM_1}\right|$ is the distance between antenna $A_1$ and projection point $M_1$.

In the same way as obtaining $\left|\overrightarrow{A_1(t_1)M_1}\right|$ and $\left|\overrightarrow{T_aM_1}\right|$, tag $T_a$, $R_2$, $E_2$ and $J_2$ forms a plane, and $\left|\overrightarrow{A_2(t_1)M_2}\right|$ and $\left|\overrightarrow{T_aM_2}\right|$ can be obtained, where $\left|\overrightarrow{A_2(t_1)M_2}\right| = x_{a,2}$. Now the $x_{a,1}$ and $x_{a,2}$ obtained by the two antennas, respectively, are the estimated position of the tag $T_a$ on the $x$-axis, and the mean value processing is

$$x_a = \frac{1}{2}(x_{a,1} + x_{a,2}) \tag{30}$$

where $\left|\overrightarrow{A_2(t_1)M_2}\right| = x_{a,2}$, $x_a$ is the real $x$-axis position estimate of tag $T_a$.

Let the line where $M_1$ and $M_2$ are projected to the $x$-axis intersect at point $o_M$, take the line where $M_1$ and $M_2$ are located as the $z_M$ axis, and the line parallel to the $y$ axis through the point $o_M$ is the $y_M$ axis. Therefore, the $y_M o_M z_M$ plane is shown in Figure 8.

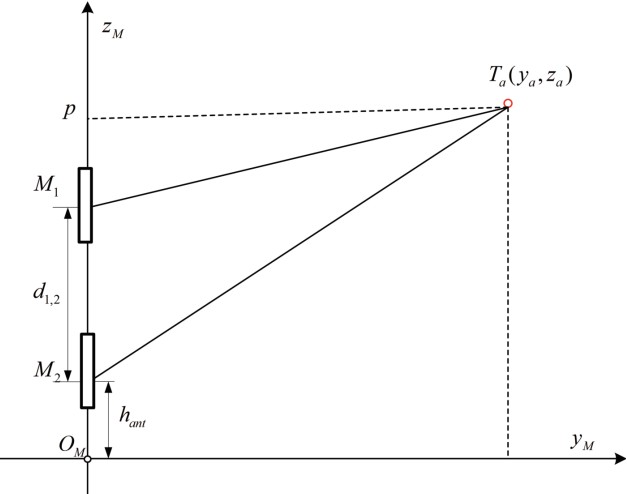

**Figure 8.** Calibration algorithm yoz profile.

As shown in Figure 8, the projection point of tag $T_a$ on the $z_M$ axis at this time is denoted as P and $M_2$ forming a right triangle with tag $T_a$, and P and $M_1$ forming a right triangle with tag $T_a$, so it can be obtained

$$\begin{cases} \left|\overrightarrow{T_aM_1}\right| = \sqrt{\left|\overrightarrow{PM_1}\right|^2 + y_a^2} \\ \left|\overrightarrow{T_aM_2}\right| = \sqrt{\left(\left|\overrightarrow{PM_1}\right| + d_{1,2}\right)^2 + y_a^2} \end{cases} \tag{31}$$

where $\left|\overrightarrow{PM_1}\right|$ is the distance between P and $M_1$, which can be solved by

$$\left|\overrightarrow{PM_1}\right| = \frac{\left|\overrightarrow{T_aM_2}\right|^2 - \left|\overrightarrow{T_aM_1}\right|^2 - d_{1,2}^2}{2d_{1,2}} \tag{32}$$

According to Equation (32), it can be obtained according to the position relation in Figure 8

$$\begin{cases} y_a = \sqrt{\left|\overrightarrow{T_aM_1}\right|^2 - \left|\overrightarrow{PM_1}\right|^2} \\ z_a = \left|\overrightarrow{PM_1}\right| + d_{1,2} + h_{ant} \end{cases} \tag{33}$$

By combining Equations (30) and (33), the three-dimensional coordinates $(x_a, y_a, z_a)$ of the tag $T_a$ estimated based on the calibration method proposed in this paper can be obtained.

The $(x_a, y_a, z_a)$ obtained above are the three-dimensional coordinates of tag $T_a$ obtained by the calibration algorithm at time $t_n$, $t_{n+w}$ and $t_{n+2w}$. Since the time length is $N = 3w$, $w$ possible coordinate positions can be solved in the 3D position calibration stage.

Directly averaging the $w$ positions usually does not achieve the best results. Therefore, weight is introduced to optimize the positioning effect of the algorithm further. Take the motion direction of the reader, i.e., the inverse of the variance of the estimated value of the x-dimension $\frac{1}{Var(x_a)}$ as the weight to further calibrate the x-dimension coordinates, and obtain

$$x_{a,fin} = \sum_{n=1}^{w} x_a(t_n) \left( \frac{\frac{1}{Var(x_a(t_n))}}{\sum_{n=1}^{w} \frac{1}{Var(x_a(t_n))}} \right) \tag{34}$$

Similar to the above Equation (23) to Equation (34), the x-dimension coordinate of the updated tag $T_b$ is $x_{b,fin}$. Therefore, the x-dimension coordinate of the target to be tested can be updated to

$$\hat{x}_{Tc} = \frac{1}{2}(x_{a,fin} + x_{b,fin}) \tag{35}$$

At the same time, $y_a$, $y_b$ together with $z_a$ and $z_b$ can also be obtained by treating the y and z-dimension results obtained from Equation (33) in the same way as that from Equations (34) and (35) in X-dimension. Since Y and Z are processed in the same way, it will not be repeated here. At this point, the estimation of the target can be updated as $MRRDT_{final}$ and its result is $(\hat{x}_{Tc}, \hat{y}_{Tc}, \hat{z}_{Tc})$.

### 2.4. Comparison of the above Two Results

Based on the same coordinate system as shown in Figure 1 above, we set up a set of experiments to compare the results of $MRRDT_{first}$ and $MRRDT_{final}$ with the traditional MRL method [13]. Change the y, z, and x-dimension coordinates of the target, respectively, while keeping the other two coordinates unchanged. In this simulation, we divided the target location into three groups according to the changes in three dimensions, namely $Group_y\{(1.3, 0.6, 1.3), (1.3, 0.8, 1.3), (1.3, 1.0, 1.3)\}$, $Group_z\{(1.3, 0.8, 0.7), (1.3, 0.8, 1.3), (1.3, 0.8, 1.9)\}$ and $Group_x\{(0.7, 0.8, 1.3), (1.3, 0.8, 1.3), (1.9, 0.8, 1.3)\}$. To better highlight the performance of these methods, we set a lower sampling rate of 10 Sa/s in this group of comparison experiments, and other parameters are set as shown in Table 1. In addition, we obtain a set of simulation results as shown in Table 2.

**Table 1.** Parameter settings.

| Object | Setting |
|---|---|
| Area size | 4 m × 4 m × 2.5 m |
| Shelf size | 2 m × 1 m × 2 m |
| Distance between tag pairs | 5 cm |
| Distance between two antennas | 0.2 m |
| Reader frequency | 924.5 MHZ |
| Sampling rate | 100 Sa/s |
| 90% Gaussian noise | 6 dBm |
| 10% Gaussian noise | 15 dBm |
| Robot moving speed | 0.2 m/s |

First, we analyze the positioning results in the *x*-axis position. As can be seen from Table 2, the $MRRDT_{final}$ method provides more accurate length estimation compared to the $MRRDT_{first}$ and MRL methods. Second, for the results of the y and *z*-axis position, the $MRRDT_{first}$ method proposed in this paper achieves significant performance improvements compared with the other two methods.

As a result, based on the above analysis of the simulation results in Table 2, the utilization of the $MRRDT_{final}$ method yields the most accurate estimation of the *x*-axis

position, while the implementation of the MRRDT$_{first}$ method provides the most precise information regarding the *y*-axis and *z*-axis position. Therefore, this paper estimated using MRRDT$_{final}$ in dimension x and MRRDT$_{first}$ in dimension y and z, i.e., the paper used $(\hat{x}_{Tc}, \hat{y}, \hat{z})$ as the final output result of the shelf positioning system (It is worth noting that if the robot does not reach the RSS peak in the moving process, the three-dimensional position obtained by position calibration in this paper is more accurate. However, after the robot achieves the RSS peak of the target tag, the accuracy of the position obtained by rough estimation in y and z dimensions is improved). The average error of the estimation of the target by the MRL method and the proposed MRRDT algorithm is shown in Figure 9.

**Table 2.** Comparison of position errors.

| Group | Targets (cm) | Position $Error(x, y, z)$ of MRL in [13] (cm) | Position $Error(x, y, z)$ of MRRDT$_{first}$ (cm) | Position $Error(x, y, z)$ of MRRDT$_{final}$ (cm) |
|---|---|---|---|---|
| *Group$_y$* | (1.3, 0.6, 1.3) | 3.34, 45.69, 68.86 | 14.47, **2.67**, **0.05** | **3.16**, 42.52, 62.23 |
| | (1.3, 0.8, 1.3) | 3.30, 36.07, 62.85 | 15.19, **2.20**, **0.06** | **3.20**, 34.50, 55.48 |
| | (1.3, 1.0, 1.3) | 3.10, 28.75, 56.15 | 16.23, **2.01**, **0.06** | **3.17**, 28.16, 48.92 |
| *Group$_z$* | (1.3, 0.8, 0.7) | 1.48, 7.30, 14.12 | 12.37, **1.36**, **0.05** | **0.91**, 5.32, 11.15 |
| | (1.3, 0.8, 1.3) | 3.38, 36.31, 62.71 | 15.45, **2.28**, **0.06** | **3.29**, 34.51, 55.46 |
| | (1.3, 0.8, 1.9) | 4.92, 66.92, 133.99 | 18.90, **3.59**, **0.07** | **6.45**, 69.44, 122.55 |
| *Group$_x$* | (0.7, 0.8, 1.3) | 6.74, 25.39, 42.53 | 14.93, **2.15**, **0.07** | **4.70**, 27.97, 49.03 |
| | (1.3, 0.8, 1.3) | 3.34, 36.03, 63.64 | 14.81, **2.10**, **0.06** | **3.10**, 34.41, 56.03 |
| | (1.9, 0.8, 1.3) | 9.39, 29.22, 40.17 | 15.80, **2.30**, **0.06** | **9.09**, 30.64, 31.97 |

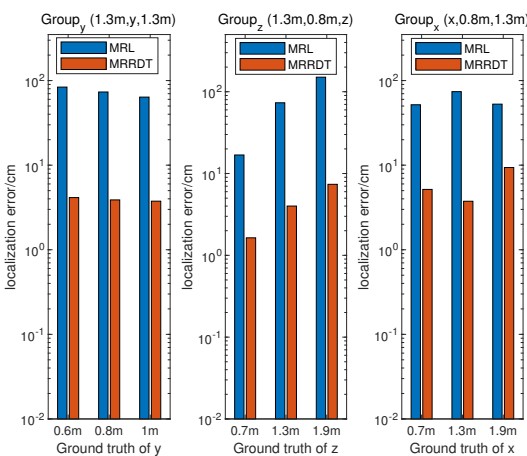

**Figure 9.** Three sets of target positioning errors.

From Figure 9, we can see that when only the depth, height, and length information of the target on the shelf are altered while keeping the coordinates of the target unchanged in the other two dimensions, the MRRDT method achieves centimeter-level 3-D positioning accuracy, while the MRT method only achieves sub-meter-level 3-D positioning accuracy.

## 3. Simulation and Analysis

To verify the performance of the proposed MRRDT algorithm, a simulation test was carried out in the range of 4 m × 4 m × 2.5 m, with the shelf length 2 m, height 2 m, and depth 1 m. The placement of the target to be tested on the shelf is shown in Figure 1. In addition, other default parameters are set as shown in Table 1. The 18 targets are placed in three rows and six columns. All objects are placed in this distribution in general, but the specific locations are obtained randomly on this basis. Each target is attached with two tags, top and bottom, a total of 18 groups of tag pairs, and the spacing of tag pairs is 5 cm (If there is no other explanation, the positioning effect of all tags on the shelves is taken as the positioning effect of the system in this paper). In addition, according to the work in [29],

the similar noise model we adopted contains two different Gaussian noises accounting for 90% and 10%, respectively. This paper does not simulate the effect of multipath noise because we can perform positioning based on LOS paths only according to the relevant work in [22].

To evaluate the positioning performance of the proposed algorithm, the root mean square error (RMSE) of positioning was used to evaluate

$$RMSE = \sqrt{\frac{1}{m}E\left[\|P - \hat{P}\|_F^2\right]} \tag{36}$$

where $P$ sequence is the real position of $m$ targets to be measured in this system and $P = [P_1, P_2, \ldots, P_m]$. The $\hat{P}$ sequence is the estimated position of $m$ targets to be measured in this system, and $\hat{P} = [\hat{P}_1, \hat{P}_2, \ldots, \hat{P}_m]$.

### 3.1. The Positioning Performance of Targets

Figure 10 shows the cumulative distribution function (CDF) curve of the three-dimensional position positioning error of the shelf target by the proposed method and the MRL algorithm [13]. As can be seen from this figure, the average error of the proposed algorithm in the *x*-axis, *y*-axis, and *z*-axis is 0.69 cm, 1.02 cm, and 0.05 cm, respectively, and the three-dimensional average error of the shelf system is 1.34 cm. The positioning performance of the *z*-axis is the best, followed by the *x*-axis and *y*-axis. The performance of the MRL algorithm in x-dimension is close to that of the proposed algorithm and even slightly better than the proposed algorithm in a certain range. However, in the y-dimension and z-dimension, even the overall positioning performance of the target to be tested is inferior to the proposed algorithm.

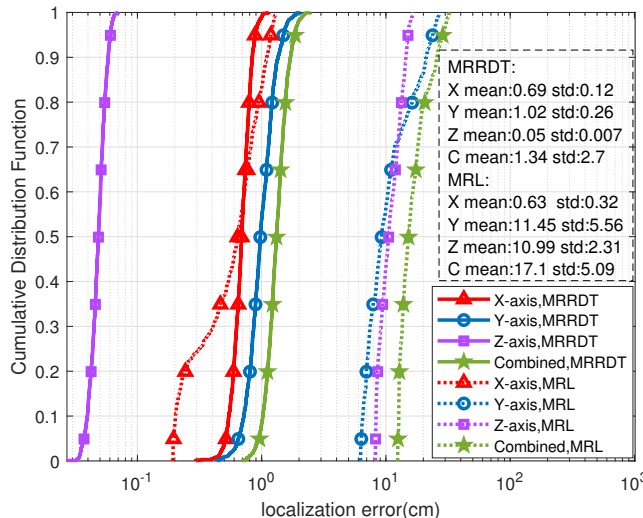

**Figure 10.** Three-dimensional results of two methods for locating targets.

### 3.2. Analysis of Other Influencing Factors

3.2.1. Influence of Different Tags on Spacing and Positioning Performance

To estimate other factors affecting the positioning accuracy of the target to be positioned, different spacing distances between the tag pairs were set to observe the influence of the spacing between the tag pairs on the 3D positioning accuracy of the target. Figure 11 illustrates the 3D root mean square positioning error diagram of the shelf target with tag pairs with different spacing.

It can be seen from this figure that the change in the spacing between the tag pairs has little influence on the positioning accuracy of the x and y dimensions of the measured target. However, when the interval between tag pairs changed from 2 cm to 8 cm, the z-dimension root mean square positioning error of the target to be measured decreased significantly. In

addition, the positioning accuracy of the z-dimension has a good distribution in the total positioning accuracy of the measured target.

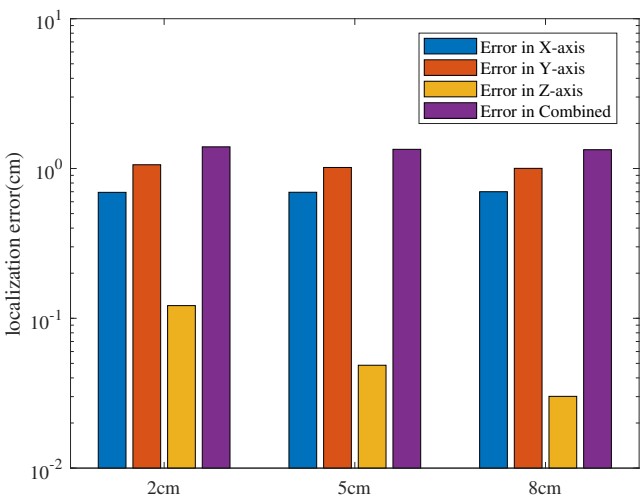

**Figure 11.** Influence of spacing between tag pairs on positioning accuracy.

3.2.2. Influence of Different Antenna Intervals on Positioning Performance

In addition, to estimate the influence of the spacing between antennas on positioning accuracy, the height of the lower antenna is fixed at 0.3 m, and the height of the upper antenna is determined by setting different antenna spacing distances, to observe the influence of different antenna spacing on the positioning accuracy of the positioning algorithm in this paper. Figure 12 shows the cumulative distribution function (CDF) curve of shelf target positioning errors corresponding to different antenna intervals.

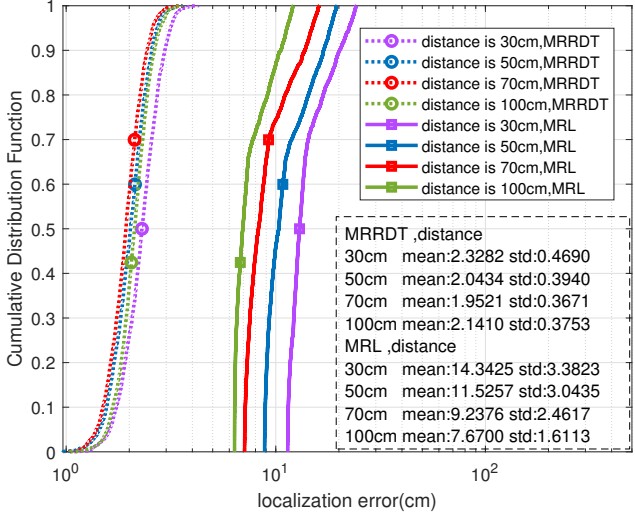

**Figure 12.** Influence of MRRDT antenna interval on positioning accuracy.

As can be seen from Figure 12, the proposed algorithm has better positioning performance than the MRL algorithm [13] when the antenna has the same spacing distance. The positioning error of the algorithm proposed in this paper generally presents a downward trend as the interval increases. When the interval increases to 70 cm, the average error remains at about 2 cm. Of course, the positioning error of the MRL algorithm also continues to decrease. The performance of the proposed algorithm is superior to that of the MRL algorithm, but irregularities are found in the antenna spacing, and the antenna spacing will be further subdivided into the following sections.

### 3.2.3. Influence of Different Sampling Rate and Antenna Interval on Positioning Performance

To discuss the influence of different antenna sampling rates and antenna spacing on positioning accuracy, the lower antenna height is fixed at 0.3 m, and the upper antenna height is determined according to the different antenna spacing distances used. In the same sampling rate environment, different antenna intervals are set to determine the height of the upper antenna to observe the influence of different antenna intervals on the positioning accuracy of the positioning algorithm in this paper. In addition, different antenna sampling rates are set to observe the influence of different sampling rates on positioning performance. Figure 13 shows the simulation results.

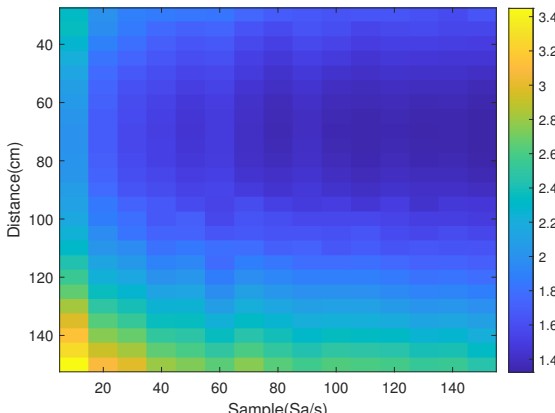

**Figure 13.** Influence of MRRDT antenna interval and sampling rate on positioning accuracy.

As can be seen from this figure, the larger the sampling rate is, the smaller the positioning error will be under the condition of the same airline interval. However, when the sampling rate is fixed and the antenna interval is less than 70 cm, the system positioning error decreases with the increase of the antenna interval. However, when the antenna interval is larger than 80 cm, the system positioning error first increases with the increase of the antenna interval. Meanwhile, the results of this figure satisfy the results shown in Figure 12.

### 3.2.4. Influence of Different Speed and Noise on Positioning Performance

To discuss the influence of robot moving speed and environmental noise changes on positioning accuracy, environmental noise is Gaussian white noise with a mean value of 0 and standard deviation $\sigma$. Then, the root mean square error of the positioning results of the robot with different moving speeds and environmental noise is calculated. Figure 14 shows the root mean square error curve of the algorithm in [13] and the algorithm in this paper under two different noise environments.

As can be seen from Figure 14, both algorithms will increase the positioning error as the robot's moving speed increases. In the same noise environment, when the robot's moving speed increases from 0.1 m/s to 1.5 m/s, the proposed algorithm has better positioning performance than the MRL algorithm. However, when the robot's moving speed is greater than 0.9 m/s, the positioning performance of the algorithm in this paper rapidly deteriorated due to the change in the robot's moving speed, while the positioning performance of the MRL algorithm is stable when the robot's moving speed is less than 0.9 m/s. Finally, the performance of the two algorithms in the Gaussian noise environment with a standard deviation $\sigma$ of 0.2 is weaker than that in the noise environment with $\sigma$ of 0.1. However, the algorithm in this paper still maintains good positioning performance in a certain speed range under the condition that the noise increases.

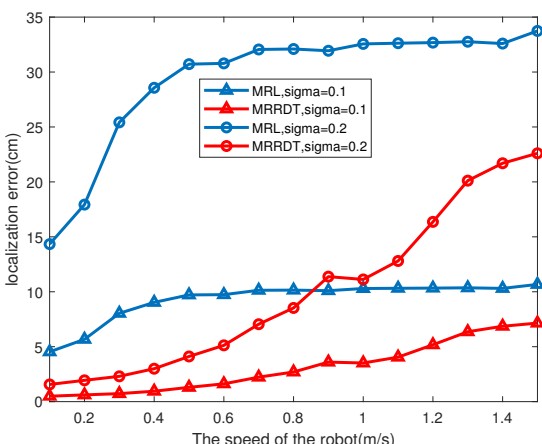

**Figure 14.** Influence of robot moving speed and ambient noise on positioning accuracy.

## 4. Conclusions

This paper proposes an MRRDT algorithm based on mobile double antennas and double tags. The RSS and phase information of the signal reflected by the tag on the target object is collected by the robot in the process of moving. First, the initial value of the x-dimension of the target is estimated using the peak signal of RSS, and the y and z-dimensions of the target are estimated using the double-tags phase model. Then, the 3D MRL is used to calibrate the rough estimate of the target, and the 3D position information of the target is obtained. Simulation results show that the proposed MRRDT system is 90% more accurate than the MRL method [13]. Based on obtaining centimeter-level positioning accuracy, it can effectively reduce deployment costs and improve equipment utilization. Due to the limited range of normal work of the RFID system, the double tags may fall off the tag, and the distance between the tag pairs will lead to the failure to match the smaller target to be tested ( if directly reduced, it will cause strong electromagnetic interference, resulting in reduced positioning accuracy ) and other problems. Therefore, the future research direction will be to deal with the situation that the target to be tested is not in the scope of the positioning system, the tag is damaged or missing to a certain tag, and the adaptive positioning of the tag to the target to be tested with different spacing.

**Author Contributions:** Conceptualization, Y.X. and T.G.; Methodology, Y.X.; Software, T.G.; Formal analysis, Y.X.; Writing—original draft, Y.X. and T.G.; Project administration, Y.X., D.Z., Y.Z. and H.H.; Funding acquisition, Y.X. All authors have read and agreed to the published version of the manuscript.

**Funding:** This work was supported by the National Natural Science Foundation of China (NSFC) under Grant 62001238 and in part by the Open Research Fund of National Mobile Communications Research Laboratory, Southeast University under Grant 2022D11.

**Data Availability Statement:** Not applicable.

**Acknowledgments:** We thank the Editors and reviewers for their support and comments.

**Conflicts of Interest:** The authors declare no conflict of interest.

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
