# Peer review of "A High-Precision 3D Target Perception Algorithm Based on a Mobile RFID Reader and Double Tags"

_remotesensing, doi:10.3390/rs15153914_

Round 1
Reviewer 1 Report
This paper proposed a three-dimensional (3D) target positioning algorithm based on a mobile RFID reader and two tags. The RSS and phase of backscattered signal are used to estimate an initial 3D position of the target. Then the above initial estimated position is corrected by the proposed calibration algorithm. The method proposed in this paper is reasonable. However, there are still some minor issues.
(1) What is the shape of the cuboid above or behind the yellow target in Figure 1 on the third page, and if it is meaningless, can it be deleted?
(2) The results of the different methods in Figure 3 on page 5 can be distinguished by different linetypes.
(3) What does the symbol "ta3/ta4" in line 172 on page 6 mean, and whether it should be "tb1/tb2"?
(4) What is the meaning of k in equation (7) on page 7, please explain.
(5) The font of i=1,2 in line 125 needs to be modified.
(6) In line 203, Φ^(u^' ) in equation (11) = [φ ̃_(a,1)^(u^' ),φ ̃_(b,1)^(u^' ),φ ̃_(a,2)^(u^' ),φ ̃_(b,2)^(u^' )], symbol ~ should be deleted.
(7) In line 179, the meaning of the symbol Φ Ìƒ^u in Equation (6) is not defined.
Please unify the format of the literature in the references.
No comments here
Reviewer 2 Report
The paper proposes a 3D target perception algorithm, which uses a mobile radio frequency identification reader and dual tags. The statement of the problem seems to be relevant, corresponds to the field of application. The article is pleasant to read, a lot of analysis and research, mathematical and experimental material. The strength of the article is the good structure of the material as well as the mathematical apparatus.
In terms of the content of the article, there are statements that do not disclose some issues. There are also statements that are difficult to agree with or require additional explanation. The above mentioned are described below.
1. Introduction
The introduction section begins with the relevance of RFID technology, the authors describe a typical RFID system, demonstrating its widespread use in manufacturing and households. The authors propose to consider the problem of achieving cost-effective and highly accurate 3D positioning in target perception in radio frequency identification research. Existing methods of positioning are given, the advantages and disadvantages of the methods, their scope of application are highlighted. However, among the list of sources used, mostly only Chinese scientific materials. You need to expand the geography of the studied literature. I recommend studying the following articles, which also use radio-frequency identification to improve conveyor lines, and in article â„–2 the authors suggest registering objects using two-step identification, which is close to your research, where you use two-step positioning of objects:
1. M. C. Caccami, S. Amendola and C. Occhiuzzi, "Method and system for reading RFID tags embedded into tires on conveyors," 2019 IEEE International Conference on RFID Technology and Applications (RFID-TA), Pisa, Italy, 2019, pp. 141-144, DOI: 10.1109/RFID-TA.2019.8892245.
2. A. Badriev, I. Makarova and P. Buyvol, "The RFID system for accounting and control of truck tires with two-step identification: a case study," 2020 13th International Conference on Developments in eSystems Engineering (DeSE), Liverpool, United Kingdom, 2020, pp. 100-104, DOI: 10.1109/DeSE51703.2020.9450743.
The introduction section ends with a logical presentation of the content of the sections, and the scientific novelty is sufficiently clear and essentially stated.
2. System architecture
In the second section, the authors describe in detail the model of perception of information about the location of objects. However, the section turns out to be very small, which is incongruous in comparison to other sections. Here, it is recommended that the authors consider merging this section with the third section and give the new section a common name.
3. RFID-based Target Perception Algorithm
In the third section, the authors reveal the operation of the RFID-based target perception algorithm. The target perception algorithm is divided into two steps: a rough estimation of the three-dimensional target position and an x-dimensional position calibration. These two steps are accompanied by the description of a mathematical model, which eventually allows the construction of a phase model of the target with double tags. Because of the interference and noise generated by the measurements, the authors suggest that calibration be performed to refine the location of the objects. However, the authors do not note what limitations exist for the mathematical model? Under what conditions is it not applicable? The authors further compare the results that the use of the MRT method provides accurate information about the length of targets on the rack. However, due to significant errors in the y and z measurements obtained by the MRT method, the accuracy of 3-D target positioning is significantly reduced. The 3-D accuracy of the target positioning is significantly reduced. The 3-D position obtained with the position calibration in this article is more accurate.
4. Simulation and Analysis
In the fourth section, to verify the performance of the proposed MRRDT algorithm, the authors conducted simulation tests. The authors quite clearly with the help of dependencies demonstrate the superiority of the proposed algorithm in comparison with others. Next, the authors analyze the influence of other factors. For example, the effect of tags on intervals and positioning, antenna intervals on positioning characteristics, sampling rate and antenna interval on positioning characteristics, speeds and noise on positioning characteristics.
5. Conclusion
In the fifth section, the authors present the results of the research and also describe the future direction of the research. In this section, the authors are encouraged to add the results of a comparison on the accuracy of the algorithm, by how much more accurate the new algorithm is than other known algorithms.
Reviewer 3 Report
Dear author,
After reviewing your manuscript, please find my suggestions. The english is correct, and the study is detail. However, there are three major issues that have to be solved before publication, in order to make it usable by the scientific community :
1. The information must be more concise. The information is cluttered and makes it difficult to understand, while the theory behind is rather simple. You could reduce its size by 50% at least, increasing the concision and readability. Examples:
(a) most of the content is repeated in formulas 6, 9, 11, 13, 14;
(b) there is a strange way to use indices make it simpler;
(c) formula 28 has too much redundant information;
(d) In essence the figures 9, 10, 11, 12, 13, and the table 1, present the same information, and should be synthetized to convey the important meaning only.
I would suggest you to read one of those books to help improving the writing concision :
- Lindsay, D., 2020. Scientific Writing = Thinking in Words, 2e édition. ed. CSIRO PUBLISHING.
- Zaumanis, D.M., 2021. Write an impactful research paper: A scientific writing technique that will shape your academic career. Independently published, Las Vegas.
2. The literature study must be completed. The statement about what is new in your study seems untrue to me. In my opinion, you should at least comment the two latest major reviews on the domain, to better identify the novelty you are bringing :
- Motroni, A., Buffi, A., Nepa, P., 2021. A Survey on Indoor Vehicle Localization Through RFID Technology. IEEE Access 9, 17921–17942. https://doi.org/10.1109/ACCESS.2021.3052316
- Xu, J., Li, Z., Zhang, K., Yang, J., Gao, N., Zhang, Z., Meng, Z., 2023. The Principle, Methods and Recent Progress in RFID Positioning Techniques: A Review. IEEE Journal of Radio Frequency Identification 7, 50–63. https://doi.org/10.1109/JRFID.2022.3233855
A few well cited studies (there are others!) that are related to yours:
- Xiao, F., Wang, Z., Ye, N., Wang, R., Li, X.-Y., 2018. One More Tag Enables Fine-Grained RFID Localization and Tracking. IEEE/ACM Trans. Netw. 26, 161–174. https://doi.org/10.1109/TNET.2017.2766526 - Wu, H., Tao, B., Gong, Z., Yin, Z., Ding, H., 2019. A Fast UHF RFID Localization Method Using Unwrapped Phase-Position Model. IEEE Transactions on Automation Science and Engineering 16, 1698–1707. https://doi.org/10.1109/TASE.2019.2895104
3. Please detail your simulation (model or experiment, it is unclear) . If your simulation is not right, then the whole point of the article does not stand. For example, multipathing is the main source of inaccuracy. How is it taken into account in your simulation ?
Then there are details, that you can find in my manually annotated pdf attached.
The english language is correct and understandable (there was just a few sentences that I could not understand). However, as stated in the general comments, the concision must be improved.
Round 2
Reviewer 2 Report
Good job, recommended for publication.